# The Influence of Health Behavior Theory on Implementation Practice and Science: Brief Review and Commentary

**DOI:** 10.3390/pharmacy10050115

**Published:** 2022-09-18

**Authors:** Anne E. Sales, Stacy L. Farr, John A. Spertus

**Affiliations:** 1Sinclair School of Nursing, Department of Family and Community Medicine, School of Medicine, University of Missouri, Columbia, MO 65211, USA; 2Center for Clinical Management Research, VA Ann Arbor Healthcare System, Ann Arbor, MI 48105, USA; 3Healthcare Institute for Innovations in Quality, University of Missouri-Kansas City, Kansas City, MO 64110, USA; 4Saint Luke’s Mid America Heart Institute, Kansas City, MO 64111, USA; 5Department of Biomedical and Health Informatics, UMKC School of Medicine, Kansas City, MO 64108, USA

**Keywords:** implementation science, implementation research, implementation practice, quality improvement, health behavior change

## Abstract

As research defines new treatments and policies to improve the health of patients, an increasing challenge has been to translate these insights into routine clinical practice to benefit patients and society. An important exploration is how theories of human behavior change fit into the science of implementation and quality improvement. In this paper, we begin with a brief review of the intellectual roots of implementation science and quality improvement, followed by a discussion of how theories and principles of behavior change can inform both goals and challenges in using behavior change theories. The insights offered through health behavior change theory have led to changes in how we plan for implementation and select, develop, design and tailor implementation interventions and strategies. While the degree to which organizational and external contexts influence the behavior of providers in these organizations varies widely, some degree of context external to the individual is important and needs adequate consideration. In short, health behavior change theory is essential but not sufficient to integrate in most implementation efforts, where priority must be given to both individual factors and contexts in which individuals operate.

## 1. Introduction

As research defines new treatments to improve the survival and health status of patients, an increasing challenge lies in how to translate these insights into routine clinical practice to benefit patients and society. Implementing change and improvement in healthcare is multi-faceted, but requires behavior change of many healthcare stakeholders. Healthcare providers, leaders, and administrators, as well as payers, patients, and other professionals, all have relevant roles and actions in translating evidence into care. This paper aims to describe how theories of human behavior change play a key role in the science of implementation and quality improvement.

## 2. Methods

We begin with a brief review of the intellectual roots of implementation science and quality improvement, followed by a discussion of how theories and principles of behavior change can inform both goals and challenges in applying behavior change theories. We use the term “implementation science” when referring to the underlying science of studying changes in healthcare delivery and “implementation practice and research” when referring more generally to work being conducted. We conducted the brief review by searching for papers that cite the Theoretical Domains Framework (TDF) (described in Section 3.1) and focusing on papers that advanced the framework and created tools for use in implementation research and practice. The search was conducted by AES and covered the period from 2005, when the first TDF paper was published, through the first half of 2022. This was not intended as an exhaustive systematic review.

## 3. A Brief Review of Implementation Science and Practice

The term “implementation science” has been in general use since the mid-2000s, with more direct applicability to health care with the launch of the journal *Implementation Science* in 2006 [1]. Even though research related to implementation is conducted in many fields, including education, social services, construction, and others, in health care the term largely focuses on the implementation of evidence-based practices or policies into care.

Evidence-based practice has its roots in the evidence-based medicine movement, sparked by the work of Archie Cochrane in the 1950s [2,3]. Evidence-based medicine generally focuses on the *efficacy* of medical practice by testing the impact of a treatment when delivered with great reproducibility on an ideal population. However, when applied more broadly, outside of the setting of a randomized trial, the *effectiveness* of treatment is described by understanding the impact on outcomes in a more real-world setting with a more diverse population of patients. Differences between efficacy and effectiveness highlighted the need to select patients more reproducibly for treatment and to deliver the therapy more consistently. This sparked the movement to create and implement practice guidelines, in which the best available evidence is framed as recommendations for practice. Initial efforts at guideline implementation focused on teaching and the dissemination of the guidelines, which were often not very effective in inducing major changes in practice patterns.

Concurrent with the initial work to implement practice guidelines, research was conducted on small area variation, to understand how variable clinical practice is. This work, with a seminal paper published in 1973 in *Science* [4,5], sparked a movement within the field of health services research to better understand and explain how and why clinical practice varies as much as it does, even within relatively small geographic regions.

The field of implementation science largely grew from these two overlapping movements: one to define the appropriate practice of medicine through producing and synthesizing evidence, and the other to describe and understand how medicine is practiced, emphasizing practice variability. Through the 1990s, as the field of implementation science began to coalesce, these issues drove considerable health services research and highlighted the need for methodological rigor in learning how best to synthesize and appraise the empirical literature. This also created increasing pressure to understand how to implement practice change in health care.

The small area variation literature also prompted a separate stream of research innovation. In a widely cited paper, Donald Berwick described methods of continuous quality improvement in health care, contrasting methods used in manufacturing industries with the standards of the time in health care [6,7]. The application of quality improvement methods from manufacturing [8,9] has been widely accepted in health care organizations, as the tools are relatively easy to learn and apply. Lean methods [10,11] as well as a number of related approaches, such as Six Sigma [12] and Define, Measure, Analyze, Improve and Control (DMAIC) [13], have been applied throughout the health care industry.

Both implementation science and quality improvement practice focus on process and tool development. Implementation science has been more focused on using and developing theory than quality improvement practice, which has focused on practical applications of these methods to improve health care quality [14,15]. In implementation research, theoretical perspectives have been drawn from the social sciences, including sociology, anthropology, and economics, and more recently, from the organizational sciences and business. Increasingly, psychological theory has been incorporated into implementation research, particularly theory focused on human behavior change. Early papers from the late 1990s and early 2000s describe ways in which implementation research uses and tests these theories to effect successful implementation of evidence-based practices.

### 3.1. Behavior Change Theory in Implementation Research

The field of implementation science continues to evolve as experience and insights require more refined strategies for understanding and changing healthcare. A seminal work that brought human behavior change theory squarely into the forefront of implementation research was a 2005 publication by an influential group of implementation scientists, health psychologists, and health services researchers in the United Kingdom [16]. This paper described a lengthy process in which over 30 human behavior change theories were assembled by experts, reviewed by the interdisciplinary group of researchers, then deconstructed into specific constructs used in each of the underlying theories. These constructs were then reassembled into a “menu of constructs” or framework [17] called the Theoretical Domains Framework (TDF). Importantly, the TDF is intended to help in understanding behaviors from any of the potential adopters of behavior change, including patients, providers, or other healthcare stakeholders, both individually or as teams [18]. As a consolidated determinants framework [19], the TDF provides important information about factors that are theorized to influence the success or failure of implementation or behavior change. The TDF was refined in 2012 [20]. The motive for the TDF revisions [20] was to disperse some of the constructs in the original domains and to add new constructs. For example, the former domain “Motivation and Goals” was split into “Intentions” and “Goals” as separate domains.

In Table 1, we provide an overview comparing domains of the TDF with two other widely used determinants frameworks, the Consolidated Framework for Implementation Research (CFIR) [21] and the Tailored Implementation for Chronic Disease checklist (TICD) [22]. It is notable that the majority of domains in the TDF are included in just one domain in either the CFIR or the TICD. The number and type of constructs in the two domains in the CFIR (Characteristics of Individuals) and TICD (Individual Health Professional Factors) are different from each other, but neither include the depth or richness of the large number of constructs (over 50) in these ten or eleven domains of the TDF.

The decision to eliminate the domain “Nature of the Behaviors” from the TDF was an important change. This domain’s constructs corresponded to important domains in the CFIR and TICD, not shown in Table 1 because the domain no longer exists in the TDF. Much of the content of the former “Nature of the Behaviors” TDF domain is contained in the CFIR domain “Characteristics of the Intervention” and in the TICD domain “Guideline Factors”. Excluding this content from the revised TDF, although justified because it pertains to the context of the actual innovation or change rather than to behavior specifically, means that using the TDF alone can miss important factors.

A large number of implementation practice and research studies have used and attempted to test specific components of underlying human behavior change theories [23,24,25,26,27,28]. The Theory of Planned Behavior [29] was initially widely used [26,30,31], but after the TDF was published, attention shifted to this more inclusive framework that allows the use of concepts beyond those in the Theory of Planned Behavior. Human behavior change theories have been applied to the use of certain implementation strategies in health care, notably audit with feedback [24,32,33,34,35,36,37].

Michie continued developing approaches using the material in the TDF, notably the Behavior Change Wheel (BCW) and the Capabilities-Opportunity-Motivation-Behavior framework (COM-B) [38]. Summarizing several existing frameworks, Michie and her colleagues describe nine different intervention functions and seven policy categories which might make those interventions possible. The interventions mostly function at the level of the individual, while policy interventions are typically at the larger societal level; a component not explicitly discussed in this approach is the organizational context within which the individual works.

### 3.2. A Brief Note on the Use of Determinants Frameworks in Implementation Practice and Research

An important problem in implementation research is defining what approaches or strategies/interventions to use in trying to implement a new evidence-based practice [39]. In the absence of determinants frameworks, which catalog the factors that have been demonstrated empirically or theoretically to affect whether implementation is successful in a specific instance [19], the usual practice has been to simply make as educated a guess as possible, often without systematic effort to understand the underlying reasons why that practice is not already being used [28,40,41]. Making efforts to understand the underlying or root causes of gaps in practice, then using a theory to select, design and tailor implementation interventions or strategies, has been argued both as a way of achieving more effective implementation more often, and to build and refine theory to understand how best to achieve effective implementation. Determinants frameworks are one component of the design process. Additional components include frameworks describing implementation strategies [42,43], which can be linked to key determinants that have been assessed as influential in a specific implementation problem using logic models [44], or other approaches [45]. Michie and colleagues’ COM-B model and related approaches bundle key determinants with prescribed interventions to support implementation planning [38].

### 3.3. Behavior Change Techniques and Their Taxonomy

In further work, Michie and colleagues created the Behavior Change Technique Taxonomy (BCTT), which catalogs 93 evidence-based techniques for changing behavior, categorized into 19 hierarchically grouped domains. These can be mapped to the 14 domains of the TDF [46], enhancing the ease with which behavior change techniques can be used to design implementation strategies or interventions, although the mapping is complex.

Implementation strategies, described in both the Expert Recommendations for Implementation Change (ERIC) project [42,47] or the Effective Practice Organization Collaboration [43], are often fairly abstract and lack the detail required to deploy them operationally. One advantage of behavior change techniques is that they primarily operate at the level of the individual, or internally to the individual, offering the opportunity to specify and design strategies using behavioral techniques that can address specific, individual-level barriers.

A key advantage to behavior change techniques is the clear link to a theoretical basis for its effect, or its mechanism of action. This link has been developed further through systematic reviews and research exploring the evidence from empirical studies that have already been completed [48,49], which describe empirically derived links between behavior change techniques and the theoretical mechanisms underlying them. Recent work has focused on making a web-based tool available to assist intervention designers in developing theoretically based interventions to support behavior change, including implementation of evidence-based practices [50]. This begins to address a core problem in implementation research, which is how to find robust and accessible links between the determinant assessed as high priority, and strategies to address these determinants, particularly negative ones (barriers) [39]. Coupled with other research focused on mechanisms of action of implementation strategies [51], these efforts may help make implementation intervention design more robust, more replicable, more reliable, and ultimately more effective.

In Figure 1, we illustrate the relationships among determinants frameworks such as the TDF, strategies and behavior change techniques, and the tools that attempt to make this linkage more explicit.

### 3.4. Case Example: Audit with Feedback

Audit with feedback is a widely used strategy in implementation practice and research, which has been systematically reviewed, and its effectiveness—and the conditions under which it appears to be effective—has been assessed using meta-analysis [32]. It has been shown to be modestly effective in increasing the likelihood that a desired behavior change is achieved as part of implementation efforts. Theories underlying its use have been proposed [36,52], and theories used in designing audit with feedback interventions have been carefully studied [34,35]. Nonetheless, there has been little change in the effectiveness of audit with feedback interventions over the last decade [53], and we still have relatively little understanding of how these interventions work [54].

Part of the complexity is the multiple theories at different levels underpinning this seemingly simple intervention. We illustrate this in Figure 2, which shows how many factors in the context of a feedback intervention, in addition to the actual design of the feedback report itself, including its mode of delivery, can vary, and each of these theories requires design elements in the feedback intervention if they are assessed as important.

Specifically, the TDF and its constructs primarily apply to individual feedback, but multiple domains might pertain, including knowledge; possibly skills if skills are applicable to the specific feedback data; social/professional role and identity; beliefs about capabilities; beliefs about consequences; reinforcement; intentions; goals; memory, attention and decision processes; emotion; and behavioral regulation. The only TDF domains that explicitly include other contextual variables would be the environmental context and resources or social influences. While these constructs may be useful, there is a great deal not included in these that is included in both the CFIR and the TICD. In the CFIR, both the inner setting and outer setting domains would apply to both the internal organizational context, as well as the external context in which the organization operates. For TICD, the individual health professional factors domain (which includes multiple constructs from the TDF) would likely apply as well as: patient factors (depending on the nature of the feedback report data); professional interactions; incentives and resources; capacity for organizational change; and social, political and legal factors which primarily refer to external context.

We note the complexity of the multiple domains and constructs within each domain. As we show in Figure 1, there are relationships, albeit complex, between the determinants in frameworks and decisions about selecting strategies and behavior change techniques. We strongly recommend working with an expert in implementation science or practice who has expertise in using determinants frameworks to design and deploy implementation strategies and interventions.

In addition to the domains of the CFIR and TICD, which provide more detailed coverage of the internal organizational context as well as the external environments, both the CFIR and the TICD include domains that focus on the specific attributes of the innovation/intervention being implemented. This reflects the decision to remove the “Nature of the Behaviors” domain from the TDF. While the principle of focusing on aspects of behavior, and the attributes of the innovation may not directly relate to the behavior of individuals, this remains an important omission for the TDF.

For example, in work conducted in Canadian long term care facilities [55,56], we found key differences between requirements for behavior change for pain assessment compared with depression screening and screening for pressure injuries. The actual behavior in each of these is different, requiring different timeframes, actions by different professionals, and different responses, all of which had impact on response to feedback.

For example, pain assessment should be done on at least a daily basis, if not every shift, because the experience of pain can change quite rapidly. Screening for depression may not need to be done as frequently, as depression symptoms are usually relatively stable over time, and weekly or even monthly assessment may be more appropriate than daily. Assessment for signs of pressure injury should be done very frequently, generally on an hourly basis for people at high risk. The difference in cadence or timing for each of these assessments affects the likelihood of changing behavior, because there are more opportunities for behavior change and rapid assessment of change in behaviors that occur more frequently.

## 4. Discussion: Current State of the Science and Practice

Health behavior change theory has had a marked impact on implementation research and practice. It has contributed particularly to an understanding of how individuals process information, make decisions, and act on cues to change behavior. The insights offered through health behavior change theory have led to changes in how we plan for implementation and select, develop, design and tailor implementation interventions and strategies. An especially promising area is the development of online tools that summarize a huge amount of prior research in health behavior change and enable implementation practitioners and researchers to readily access recommendations about which behavior change techniques to include in implementation interventions, once determinants have been assessed. We note that discussion of conducting determinant assessment is outside the scope of this article. A good overview of one of these tools is given in Johnston et al., 2021 [50].

However, we have also pointed out some limitations in the current tools and approaches used in health behavior change when they are applied to implementation research. In particular, implementation practice and research in health care often takes place within the context of organizations whose mission is to deliver health care services, or to enhance health and wellbeing through programs such as well child immunization or other activities (e.g., gyms and other community-based organizations). While the degree to which organizational and external contexts influence the behavior of providers in these organizations varies widely, some degree of context external to the individual is important and needs adequate consideration. Some of this is included in the External Context and Resources and Social Influence domains of the TDF, but this content is limited in its granularity and nuance. We recommend that use of a consolidated determinants framework like the TICD, which includes key constructs from the TDF as well as the CFIR, can address this limitation. In short, while a focus on the individual is essential in most implementation efforts, it is equally important to focus on and understand the context in which the individual operates.

## 5. Conclusions

Implementing evidence into clinical practice and society is multi-faceted and requires behavior change from multiple healthcare stakeholders. These behaviors influence how we plan, design, and customize interventions and implementation strategies and overcome barriers. While quality improvement also exists within organizational contexts and priorities outside of individual behavior change, health behavior change theory is a critical underpinning to implementation science.

## Figures and Tables

**Figure 1 pharmacy-10-00115-f001:**
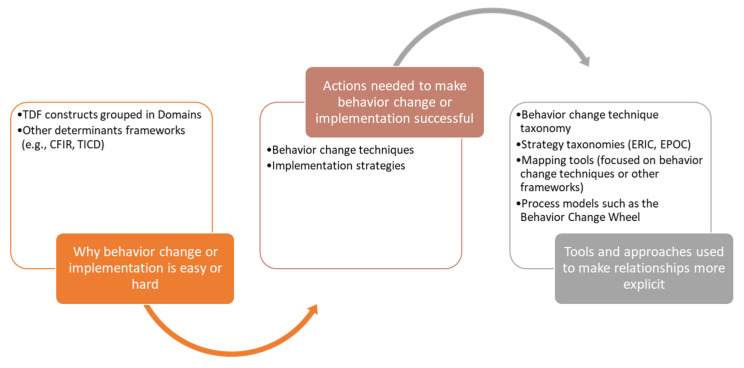
Relationships among determinants frameworks, strategies and tools.

**Figure 2 pharmacy-10-00115-f002:**
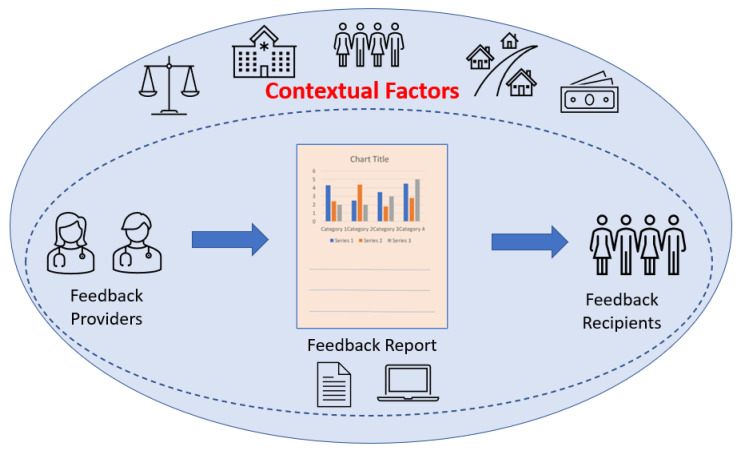
Multiple levels of elements affecting the effectiveness of a feedback report.

**Table 1 pharmacy-10-00115-t001:** Comparing Domains of the TDF, CFIR and TICD.

TDF Domain	CFIR Domain	TICD Domain
Knowledge	Characteristics of Individuals	Individual Health Professional Factors
Skills	Characteristics of Individuals	Individual Health Professional Factors
Social/Professional Role and Identity	Characteristics of Individuals	Individual Health Professional Factors
Beliefs about Capabilities	Characteristics of Individuals	Individual Health Professional Factors
Optimism	Characteristics of Individuals	Individual Health Professional Factors
Beliefs about Consequences	Characteristics of Individuals	Individual Health Professional Factors
Reinforcement	Characteristics of Individuals	Individual Health Professional Factors; Incentives and Resources
Intentions	Characteristics of Individuals	Individual Health Professional Factors
Goals	Characteristics of Individuals	Individual Health Professional Factors
Memory, Attention and Decision Processes	Characteristics of Individuals	Individual Health Professional Factors
Environmental Context and Resources	Outer Setting; Inner Setting	Professional Interactions; Capacity for Organizational Change; Patient Factors; Social, Political and Legal Factors
Social Influences	Inner Setting	Professional Interactions
Emotion	Characteristics of Individuals	Individual Health Professional Factors
Behavioral Regulation	Characteristics of Individuals; Inner Setting	Individual Health Professional Factors; Incentives and Resources; Professional Interactions

## Data Availability

Not applicable.

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
