# Peer review of "The Influence of Health Behavior Theory on Implementation Practice and Science: Brief Review and Commentary"

_pharmacy, 2022, doi:10.3390/pharmacy10050115_

Round 1

Reviewer 1 Report

Dear authors

A paper: The influence of health behavior theory on implementation 2 practice and science: Commentary

I can say that your paper is very vital,

However I see 2 problems :

Just add some sentences - Introduction (your 2 sentences are not enough)

Second problem is - and - condition sine qua non - add part - CONCLUSION

Reconsider after major revision

Reviewer 2 Report

This is a brief review and discussion on how theories and principles of behavior change can inform goals and challenges in implementation practice and science.

Title: Indicate that a brief review is conducted as well.

Introduction: Concise and clear. Please identify a clear research question or aim being posed.

Methods: How was the brief review conducted? Was there an informal search strategy or librarian involved? Is there any framework or guidance used for interpretation of key findings?

Brief Review: Very clear and well written. In Section 3.1, please clarify if the TDF domains and factors pertain to the patient. Are they assessed at the individual patient level or health care provider level or other? It is briefly addressed at the end but unclear in the second to third paragraph. In 3.4, Figure 1 could be improved by highlighting the variables discussed (e.g. mode of delivery, feedback report design). In 3.4, organizational context is briefly discussed. Consider discussing findings related to systemic factors for implementation as that may be a key point that hasn’t been elaborated on.

Discussion: Balanced interpretation and summary of findings. If possible, reiterate the online tools available that summarize health behavior change research.

Writing style: Please check the Brief Review section for minor spelling, grammar, and typing errors.

Reviewer 3 Report

Overall the manuscript describes how theories of human behaviour change into the science of implementation and quality improvement.  This an interesting and well written commentary.  A few points to consider below:

·       I believe that the ‘brief background to implementation science and practice’ starts off nicely and really sets the scene.  However, there are large sections that are quite technical and I think these sections need to be combined or condensed, with the context of healthcare at the forefront such that the paper appeals to a wider audience. E.g. lines 72-90, 92-140, 162-187, 190-230

·       I believe Figure 1 (Multiple levels of elements affecting the effectiveness of a feedback re- port) can be improved.  I’m not sure that the figure is making the point that the authors are trying to make.

Round 2

Reviewer 1 Report

Dear Authors

Now this article is much better.

I recommend to add

Maturkanič, P.; Tomanová Čergeťová, I.; Konečná, I.; Thurzo, V.; Akimjak, A.; Hlad, Ľ.; Zimny, J.; Roubalová, M.; Kurilenko, V.; Toman, M.; Petrikovič, J.; Petrikovičová, L. Well-Being in the Context of COVID-19 and Quality of Life in Czechia. Int. J. Environ. Res. Public Health 2022, 19, 7164. https://doi.org/10.3390/ijerph19127164

Budayová, Z., Pavliková, M. ., Samed Al-Adwan, A. ., & Klasnja, K. . (2022). The Impact of Modern Technologies on Life in a Pandemic Situation. Journal of Education Culture and Society, 13(1), 213–224. https://doi.org/10.15503/jecs2022.1.213.224

Hašková, A.; Šafranko, C.; Pavlíková, M.; Petrikoviˇcová, L. Application of online teaching tools and aids during corona pandemics
2020. Ad Alta 2020, 10, 106–112.

Tkáˇcová,H.; Pavlíková,M.; Tvrdon,M.; Jenisova, Z. TheUse ofMedia in the Field of Individual Responsibility for SustainableDevelopment
in Schools: A Proposal for an Approach to Learning about Sustainable Development. Sustainability 2021, 13, 4138.

Author Response

We appreciate your suggestions, but none of the papers suggested are relevant to the Commentary we are submitting. We respectfully decline to add these references.